# Design of PCF Supporting 86 OAM Modes with High Mode Quality and Low Nonlinear Coefficient

Yang Yu [1,2], Yudong Lian [1,2,3,*], Qi Hu [1,2], Luyang Xie [1,2], Jie Ding [1,2], Yulei Wang [1,2,3] and Zhiwei Lu [1,2,3]

1    Center for Advanced Laser Technology, Hebei University of Technology, Tianjin 300401, China;
     202031903055@stu.hebut.edu.cn (Y.Y.); 202131903008@stu.hebut.edu.cn (Q.H.);
     202121902027@stu.hebut.edu.cn (L.X.); dingjie@hebut.edu.cn (J.D.); wyl@hebut.edu.cn (Y.W.);
     zhiweilv@hebut.edu.cn (Z.L.)
2    Hebei Key Laboratory of Advanced Laser Technology and Equipment, Tianjin 300401, China
3    Tianjin Key Laboratory of Electronic Materials and Devices, Tianjin 300401, China
*    Correspondence: ydlian@hebut.edu.cn

**Abstract:** A unique photonic crystal fiber with square and circular air holes (SC-PCF) is designed in this research. Three layers of circular air holes and two levels of square air holes make up the fiber cladding. The finite element approach is used to simulate the fiber construction, and numerical calculations are used to examine the transmission properties in the S+C+L band. The results reveal that the SC-PCF can sustain 86 Orbital Angular Momentum (OAM) modes in the wavelength range of 1400 nm to 1700 nm (300 nm), with an effective refractive index difference (ERID) of $5.88 \times 10^{-3}$ between them, thus avoiding mode coupling. The mode purity of all modes is greater than 96% at 1550 nm, and the lowest dispersion and dispersion change are 4.939 ps/nm/km and 0.956 ps/nm/km, respectively. The confinement loss (CL) of all modes is lower than $10^{-9}$ dB/m, and the nonlinear coefficient (NC) is lower than $1.5\ \mathrm{W^{-1} \cdot km^{-1}}$ in the whole band. The proposed SC-PCF has important value in long-distance and large-capacity communication systems.

**Keywords:** optical fiber communication; optical fiber transmission characteristics; orbital angular momentum; photonic crystal fiber





## 1. Introduction

The communication capacity of single-mode fiber (SMF) is increasingly being saturated as mobile communication networks increase, and is expected to approach the Shannon Limit around 2025 [1,2]. To solve this problem, multiplexing technology is adopted, such as time-division multiplexing (TDM) [3], wavelength-division multiplexing (WDM) [4], polarization-division multiplexing (PDM) [5], and space-division multiplexing (SDM) [6], in which SDM is a very important method because SDM can enable the systems to communicate without interfering with each other, maximize the use of resources, and greatly increase the capacity of the optical communication system [7]. Orbital Angular Momentum (OAM) multiplexing, as a kind of SDM, has attracted widespread attention in the optical communication field [8–10]. OAM has the spiral wavefront phase characteristic, and the phase can be written as *exp(ilφ)*, where *l* is the angular quantum number, also referred to as the topological charge number, and *φ* is the radial quantum number. In theory, OAM has infinite topological charges due to the orthogonality between different OAM modes, which is the advantage of OAM used in SDM [11]. Since conventional optical fibers cannot transmit OAM modes, new fibers must be designed to carry them.

Photonic crystal fiber (PCF) has many excellent advantages, such as cutoff-free single-mode transmission, wide mode field area, strong birefringence, high nonlinearity, etc. [12]. These advantages can enable the fiber structure to be flexible, reduce transmission loss effectively, and enlarge the transmission bandwidth. In 2012, Yue et al. [13] first proposed a hexagonal PCF that can be used for transmitting OAM modes. Only two OAM modes can

be supported by the PCF, and the dispersion and confinement loss (CL) are relatively high. Circular PCF has the characteristics of flat dispersion and low CL, which can be used to optimize the transmission characteristics of PCF [14]. Tian et al. [15] proposed a circular PCF with flat dispersion and low CL. However, there are fewer than 26 OAM modes supported by this PCF. In order to increase the number of OAM modes and reduce the confinement loss, a PCF with square holes was proposed by Bai et al. [16]. This structure ensures a considerable difference in refractive index, and prevents high-order mode leakage by increasing the air filling rate. The experiment showed that the structure supports 46 OAM modes. Subsequently, Yang et al. [17] and Ke et al. [18] further optimized Bai's structure, and they proposed C-PCF with square air holes, which makes the number of OAM modes higher than 50. With the development of PCFs, new structures are emerging. A hollow circular PCF construction was proposed by Hong et al. [19]. The structure can support 101 OAM modes with a mode purity of more than 78.7%. Ma et al. [20] proposed a PCF that could carry 180 OAM modes while maintaining flat dispersion. The transmission bandwidth covers the C and L bands, and materials with a refractive index of 1.56 are doped into the structure. In the same year, Zhao et al. [21] proposed a PCF structure with two rings capable of 84 OAM modes to transmit. The above structures are listed in Table 1. Based on the design principles of PCF, these structures optimize the characteristics of dispersion, CL, effective mode field area (EMFA), etc., which can improve the communication capacity of optical fibers significantly.

**Table 1.** The development of PCF.

| Structure | | | | | | | | |
|---|---|---|---|---|---|---|---|---|
| Author | Yue | Tian | Bai | Yang | Ke | Hong | Ma | Zhao |
| Number of OAM Modes | 2 | <26 | 46 | >50 | >50 | 101 | 180 | 84 |
| CL (dB/m) | 0.03 | 0.003 | $10^{-9}$ | $>10^{-11}$ | $>10^{-11}$ | $<10^{-8}$ | $10^{-12} \sim 10^{-7}$ | $>10^{-13}$ |
| Reference | [12] | [15] | [16] | [17] | [18] | [19] | [20] | [21] |

In this paper, a new type of PCF with square and circular air holes (SC-PCF) is proposed. Two layers of square air holes and three layers of circular air holes make up the SC-PCF. The finite element simulation program is used to simulate the SC-PCF. The structure's transmission characteristics are investigated. According to numerical analysis, the SC-PCF supports 86 OAM modes in the S+C+L bands, and the ERID between EH and HE modes in the same OAM mode family is as high as $5.88 \times 10^{-3}$, which can prevent the coupling between adjacent vector modes effectively. The purity of all modes at 1550 nm is higher than 96%. In addition, the PCF structure has the characteristics of large EMFA, low CL and NC, and flat and low dispersion. The proposed SC-PCF is suitable for long-distance and large-capacity optical fiber communication networks.

## 2. System Structure

The superposition of adjacent vector modes generates the OAM mode. This section introduces the theory of OAM mode composition and gives the structure parameters of the proposed SC-PCF.

### 2.1. The Theory of SC-PCF Design

The modes transmitted in PCF contain four types of vector modes, which are transverse electric wave (TE) modes, transverse magnetic wave (TM) modes, and mixed wave (HE, EH) modes. The OAM mode is formed by the conjunction of the vector modes (HE and EH)

with the same propagation constant through $\pi/2$ phase difference. Therefore, the mode dispersion generated by mode departure has no effect on the OAM mode [22].

The OAM mode in the optical fiber can be written as $OAM_{l,m}$, where $l$ ($l = \pm1, \pm2, \pm3\ldots$) denotes the topological charge, and $m$ denotes the radial mode order. The OAM mode can be obtained by vector mode superposition:

$$\left.\begin{cases} OAM^{\pm}_{\pm l,m} = HE^{even}_{l+1,m} \pm jHE^{odd}_{l+1,m} \\ OAM^{\mp}_{\pm l,m} = EH^{even}_{l-1,m} \pm jEH^{odd}_{l-1,m} \end{cases}\right\}(l > 1) \\ \left.\begin{cases} OAM^{\pm}_{\pm 1,m} = HE^{even}_{2,m} \pm jHE^{odd}_{2,m} \\ OAM^{\mp}_{\pm 1,m} = TM_{0,m} \pm jTE_{0,m} \end{cases}\right\}(l = 1) \tag{1}$$

where the superscript "$\pm$" indicates the right or left direction of the circular polarization, the subscript "$\pm$" indicates the right or left rotation direction of the helical phase wavefront, and the superscripts "*even*" and "*odd*" indicate the even and odd modes derived by linear superposition. The circular polarization direction of $OAM^{\pm}_{\mp l,1}$ is the same as the wavefront rotation, but the polarization direction of $OAM^{\pm}_{\mp l,1}$ is opposite to the wavefront rotation [23, 24]. The OAM beam has four representations when $|l|>1$, due to the varied orientations of circular polarization and spiral phase wavefront rotation, as shown in Figure 1, and the specific formula is presented in Equation (1).

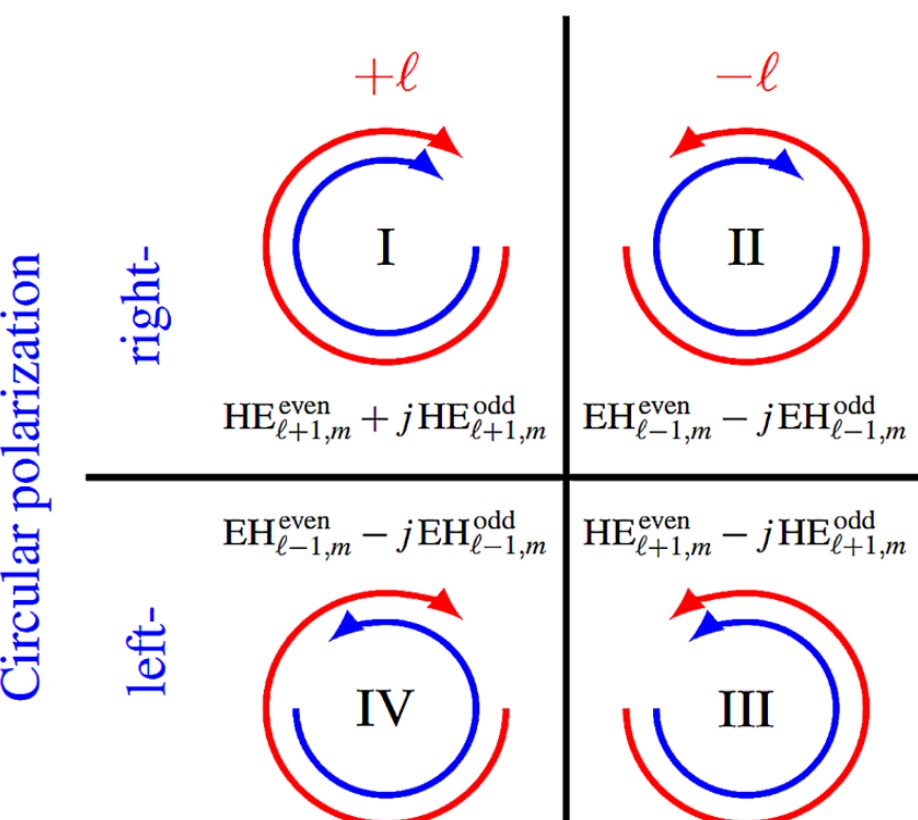

**Figure 1.** Four OAM beam representations with the same topological charge [23].

### 2.2. SC-PCF Structure

To design a PCF supporting OAM mode, the following requirements should be met: (a) the ERID between EH and HE modes in the same OAM mode family should be more than $1 \times 10^{-4}$ to avoid coupling between neighboring modes; (b) more OAM modes should be supported by PCF in a larger bandwidth range; (c) the designed PCF should have low

CL and NC, large EMFA, high mode purity, and flat dispersion; (d) the fiber structure should match the annular phase of the OAM mode field distribution [25].

Based on the above requirements, this paper proposes an SC-PCF whose cross-sectional diagram is shown in Figure 2. The cladding is designed as a combination of square and circular air holes. By adjusting the size of the air holes appropriately, the ERID between the cladding and the core can be improved without precise doping of the high refractive index material, preventing higher-order mode leakage [16,26].

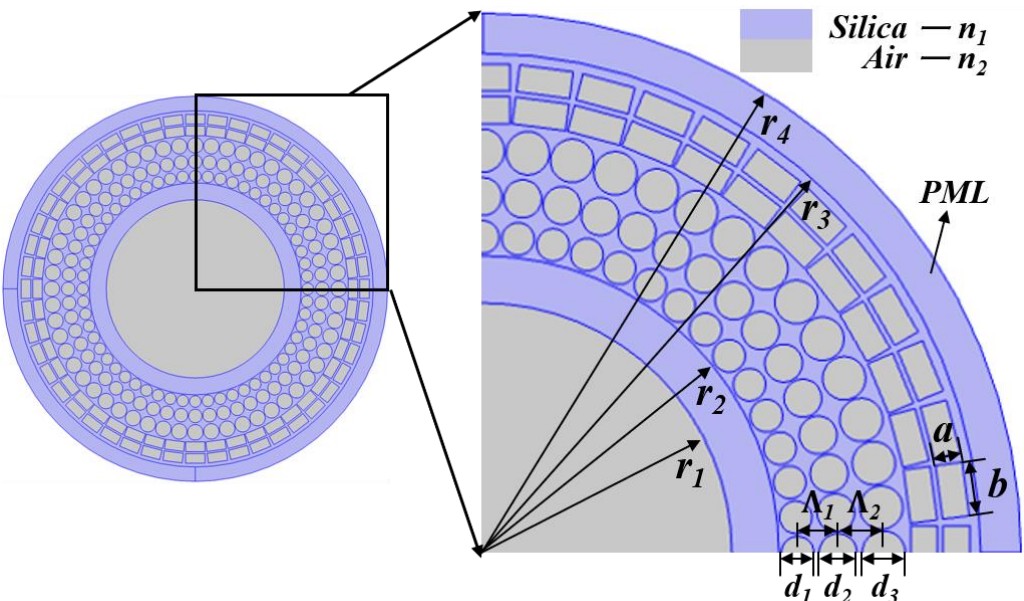

**Figure 2.** Cross-sectional diagram of a photonic crystal fiber.

With the development of optical fiber technology, the production of square holes can be realized through sol, gel, casting, and other methods [27]. Jiang et al. [28] proposed a square hole PCF with high birefringence transmission characteristics. Bai et al. [16] proposed a square air hole PCF supporting OAM mode transmission. By using Bessel polygons as air hole cladding, Hassan [29] and Kabir [30] were able to achieve PCFs with low CL and flat dispersion properties. They developed a fused silica-based fiber structure with Bessel polygon air holes to increase the optical properties, based on the fiber structures of the two previous teams. This structure resembles a spider web in cross-section [31]. Vienne et al. [32] used the stack-draw method to draw a hollow-core Bragg fiber with a microstructural cladding. The total length of the fiber is over 150 m and the transmission loss is as low as 1 dB/m. Consequently, it is feasible to fabricate a PCF with square holes.

Figure 2 shows the cross-section of the SC-PCF structure. The gray area describes the air holes and the blue area represents silica. $r_1$ represents the radius of the fiber core, $r_2$ represents the inner diameter of the annular structure, $r_4$ represents the radius of SC-PCF, and the difference between $r_3$ and $r_4$ represents the thickness of the perfect match layer (PML), which can be used to simulate the absorption conditions at the boundary and calculate the transmission characteristics of the PCF. The diameters of the circular air holes on the three layers are $d_1$, $d_2$ and $d_3$, respectively. The length and width of the square air holes on the two outermost layers are a and b, and the spacing of the adjacent air holes is $\Lambda$. When designing a PCF, the number and transmission properties of OAM modes are affected by the size and number of air holes in the cladding. After numerical analysis, the optimal parameters were determined as shown in Table 2.

**Table 2.** The main parameters of SC-PCF.

| Symbols | Parameters | Symbols | Parameters |
|---------|-----------|---------|-----------|
| $r_1$ | 9.25 μm | $d_2$ | 1.4 μm |
| $r_2$ | 11 μm | $d_3$ | 1.6 μm |
| $r_3$ | 18.5 μm | $a$ | 1 μm |
| $r_4$ | 20 μm | $b$ | 2 μm |
| $n_1$ | 1.444 | $\Lambda$ | 0.2 μm |
| $n_2$ | 1 | $\Lambda_1$ | 1.5 μm |
| $d_1$ | 1.2 μm | $\Lambda_2$ | 1.7 μm |

## 3. Simulation Results and Discussion

In this section, the finite element method is adopted to carry out the numerical calculation on the designed SC-PCF, and the transmission characteristics of the OAM mode within the band range from 1400 nm to 1700 nm are analyzed, including OAM mode distribution, effective refractive index (ERI), ERID, dispersion, CL, EMFA, numerical aperture (NA), NC, and mode purity. Because of the risk of "accidental degradation", OAM mode does not enable radial orders greater than 1, and brings problems to the demultiplexing, reuse, and encoding of OAM mode [14]. Therefore, $m = 1$ is used in this paper.

### 3.1. OAM Mode Analysis

Figures 3 and 4 list the mode field distribution and the electric field intensity distribution in *Ez* direction of OAM modes, respectively. The listed modes can be perfectly confined within the ring. In Figure 3, the mode field distributions of HE and EH modes with the same order are consistent. Therefore, HE and EH modes cannot be effectively distinguished only from the mode field distributions. However, the distribution of electric field intensity between different modes is not consistent. In the *Ez* direction diagram, the red and blue areas represent electric field intensity. Figure 4 shows that the $HE_{3,1}$, $HE_{5,1}$, $HE_{10,1}$, $HE_{21,1}$ have 3, 5, 10, 21 pairs of red and blue areas, respectively, and EH mode also has the same rule. At the same time, the electric field intensity of HE mode is closer to the outside of the ring, and that of EH mode is closer to the inside of the ring [16]. Therefore, HE and EH modes can be distinguished by the red–blue pairs of *Ez* direction and the area of the electric field intensity distribution.

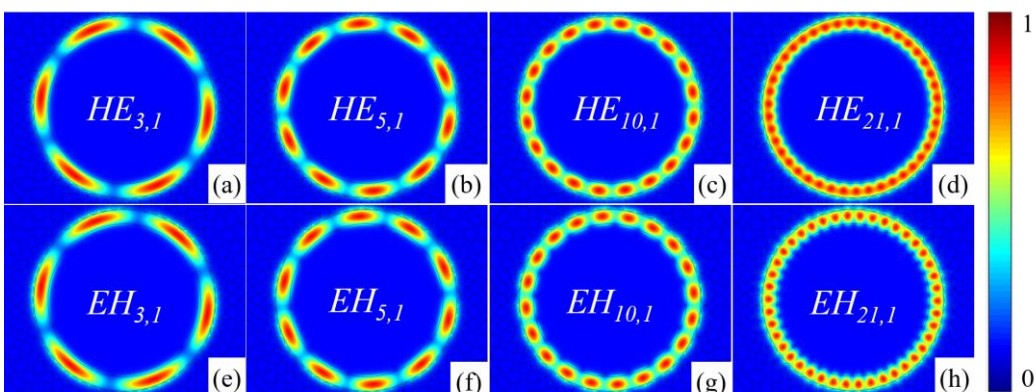

**Figure 3.** Mode field distribution diagram: (**a**) $HE_{3,1}$, (**b**) $HE_{5,1}$, (**c**) $HE_{10,1}$, (**d**) $HE_{21,1}$, (**e**) $EH_{3,1}$, (**f**) $EH_{5,1}$, (**g**) $EH_{10,1}$, (**h**) $EH_{21,1}$.

Referring to Equation (1), OAM mode can be synthesized by the superposition of the odd mode and even mode of HE mode and EH mode. Figure 5 shows the synthesis process and phase diagram of $OAM_{4,1}$, $OAM_{6,1}$, $OAM_{11,1}$, $OAM_{22,1}$ modes. The phase change of OAM mode is $2\pi l$ with the increase in topological charge number, as shown in the figure [15]. As a result, the topological charge number of the OAM mode may be clearly determined from its phase diagram.

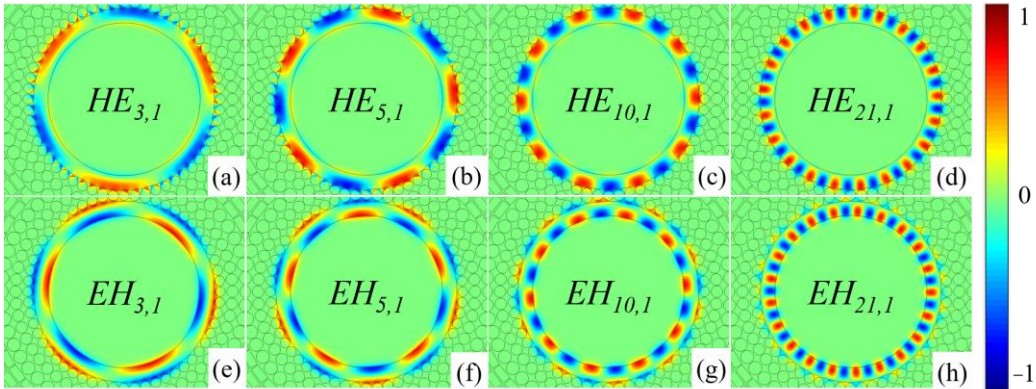

**Figure 4.** Electric field intensity distribution diagram: (**a**) HE$_{3,1}$, (**b**) HE$_{5,1}$, (**c**) HE$_{10,1}$, (**d**) HE$_{21,1}$, (**e**) EH$_{3,1}$, (**f**) EH$_{5,1}$, (**g**) EH$_{10,1}$, (**h**) EH$_{21,1}$.

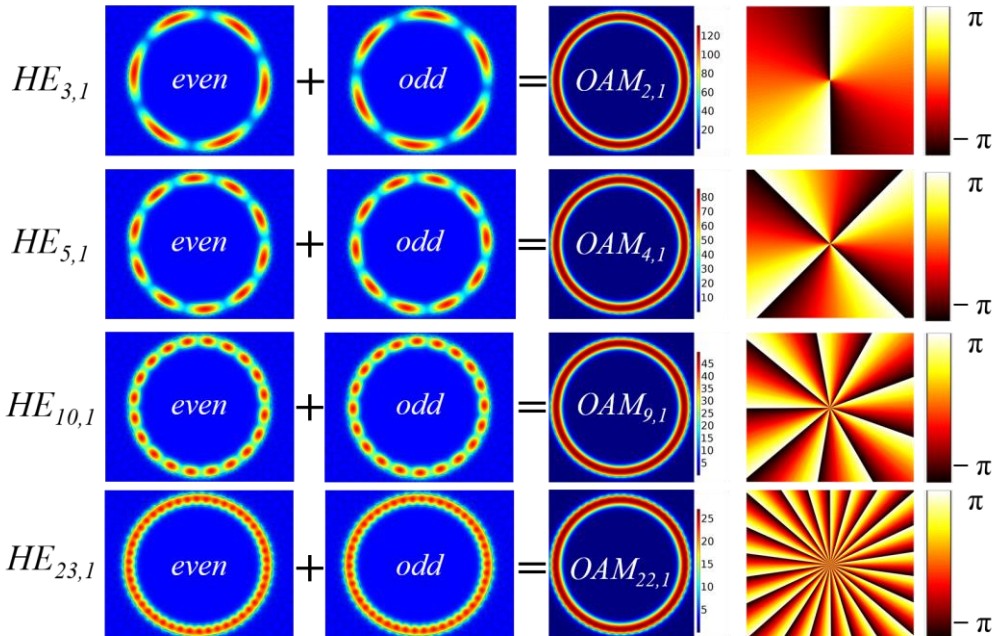

**Figure 5.** OAM$_{2,1}$, OAM$_{4,1}$, OAM$_{9,1}$, OAM$_{22,1}$ mode generation process and corresponding wavefront phase distribution diagram.

According to the composition rule of OAM modes, the SC-PCF proposed in this paper supports the topological charge *l* = 22 and can carry 86 OAM modes.

### 3.2. Effective Refractive Index (ERI) and Its Differences (ERID)

ERI (*neff*) is defined as the average refractive index weighted by light intensity distribution and can be calculated by simulation. Then, the ERID of adjacent vector modes constituting the same order OAM mode can be calculated. Different modes have a different propagation constant $\beta$. The relationship between *neff* and $\beta$ can be expressed as [33]

$$n_{eff} = \frac{\beta}{k_0} \tag{2}$$

where $k_0 = 2\pi/\lambda$, represents the wavenumber in the vacuum, and $\lambda$ is the wavelength.

Figure 6 shows the relationship between wavelength and the ERI in the wavelength range from 1400 nm to 1700 nm. The ERI decreases gradually with increasing wavelength. Meanwhile, the ERI will decrease faster with the increase in mode order, which is caused by the diffusion of the optical field into the cladding.

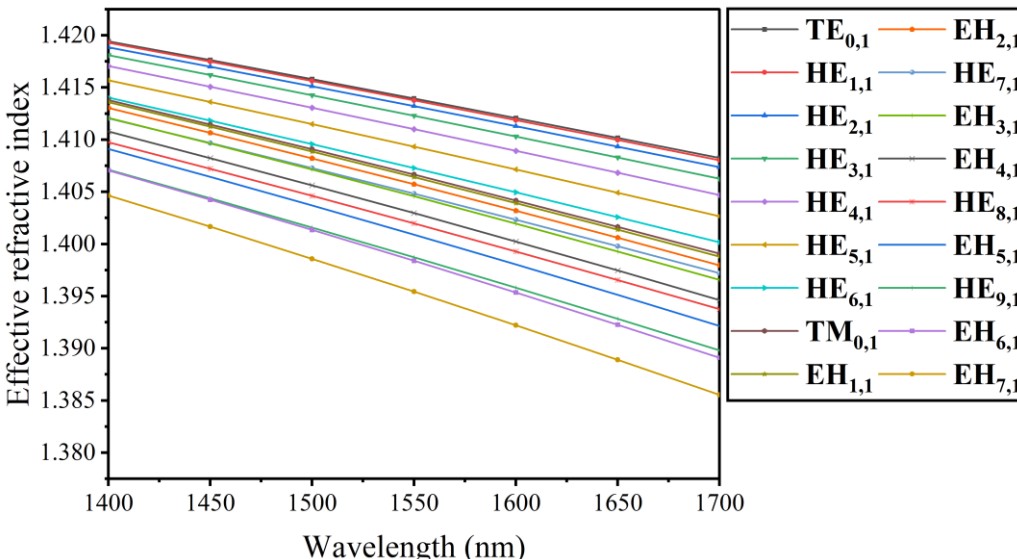

**Figure 6.** The relation between ERI and wavelength in different vector modes.

Figure 7 shows the ERID between modes supported by SC-PCF. The figure shows that the ERID increases gradually as the wavelength increases, and that the greater the order, the lower the ERID. Furthermore, all ERIDs are greater than $1 \times 10^{-4}$. At the wavelength of 1550 nm, the ERID between $HE_{3,1}$ and $EH_{1,1}$ is as high as $5.88 \times 10^{-3}$, which is much larger than the recently published reference [21]. The large ERID indicates that the fiber has great mode separation characteristics, which is of great research value in multi-channel fiber communication systems. To avoid coupling between adjacent modes, the ERID should be greater than $1 \times 10^{-4}$. The ERID can be expressed by the following equation [34]:

$$\Delta n_{eff} = \left| n_{eff_{HE_{l+1,m}}} - n_{eff_{EH_{l-1,m}}} \right| > 10^{-4} \tag{3}$$

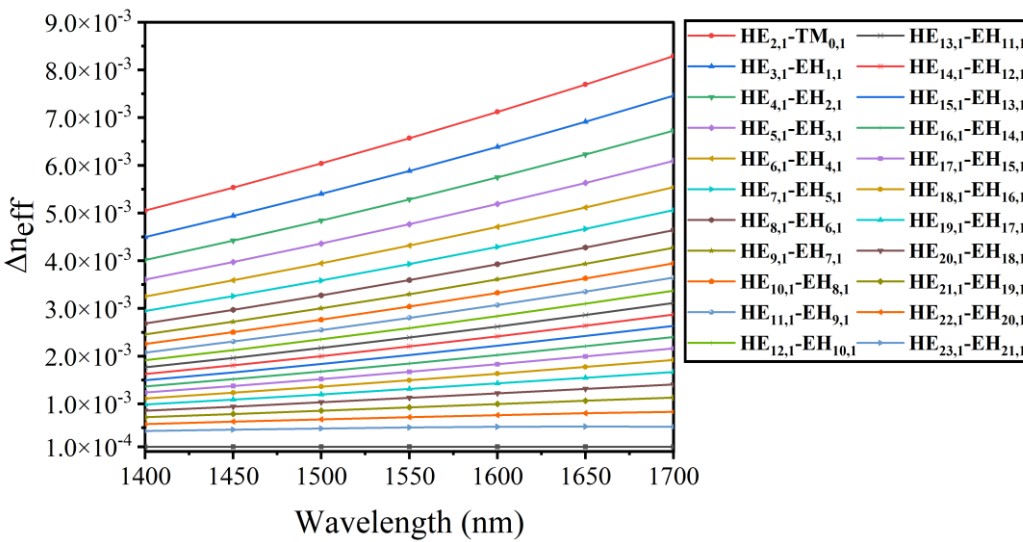

**Figure 7.** The relation between ERID and wavelength in different vector modes.

### 3.3. Dispersion Properties

Dispersion is an important transmission characteristic of the optical fiber. The dispersion will cause light pulse broadening and signal transmission distortion; thus, the optical communication transmission bandwidth is limited. The lower the dispersion, the greater the information capacity carried. Dispersion in the optical fiber includes material

dispersion and waveguide dispersion, but material dispersion is usually not considered in long-distance communication. The waveguide dispersion of SC-PCF was calculated by the following equation [35]:

$$D = -\frac{\lambda}{c}\frac{\partial^2 \left|\text{Re}\left(n_{eff}\right)\right|}{\partial \lambda^2} \qquad (4)$$

where $c = 2.9979 \times 10^8$ m/s represents the speed of light in vacuum, and Re (*neff*) is the real part of the ERI.

The dispersion of SC-PCF is shown in Figure 8. In the range from 1400 nm to 1700 nm, the dispersion of the higher-order mode increases with the increase in wavelength, whereas that of lower-order modes increases more flatly with the increase in wavelength, which is connected to the change in ERI with wavelength. The EH mode has a higher dispersion than the HE mode. In addition, the dispersion increases as the mode order increases, so the higher-order mode will cause pulse broadening, leading to the instability of the mode transmission. The minimum dispersion changes by 0.956 ps/nm/km, indicating that the dispersion changes of various modes are relatively stable. The dispersion changes of $HE_{1,1}$, $HE_{2,1}$, $HE_{3,1}$, and $HE_{4,1}$ modes are 0.973 ps/nm/km, 1.024 ps/nm/km, 1.108 ps/nm/km, and 1.228 ps/nm/km, which are far lower than the minimum dispersion value in the recently proposed reference [18,30,35]. At the wavelength of 1550 nm, the lowest dispersion is only 4.939 ps/nm/km, and the overall dispersion is less than 71.6 ps/nm/km. The dispersion is relatively flat, which is conducive to the transmission of OAM mode in the fiber.

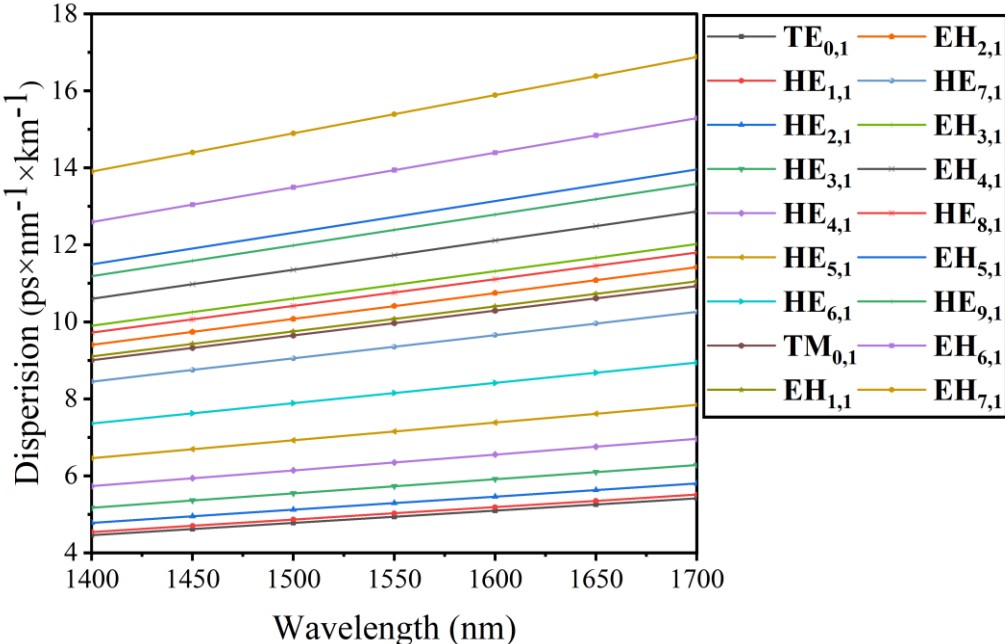

**Figure 8.** The relation between dispersion and wavelength in different vector modes.

### 3.4. Confinement Loss (CL)

CL, which means that the light field spills out from the fiber core to the cladding during transmission, is an important transmission parameter in SC-PCF, resulting in partial energy loss. This leads to a reduction in beam quality and shortens the transmission distance ultimately. By adjusting the number and diameter of air holes in the cladding, CL can be reduced effectively. CL can be calculated by the imaginary part of the ERI, which is expressed in the following form [36]:

$$CL = \frac{2\pi}{\lambda}\frac{20}{\ln 10}\text{Im}\left(n_{eff}\right)(\text{dB/m}) \qquad (5)$$

where Im (*neff*) represents the imaginary part of the ERI.

Figure 9 shows the CL distribution of different modes in the range of 1550 nm. It can be seen that the fluctuation of each mode is obvious without uniform regularity, but the overall loss is confined to the order of $10^{-8}$ dB/m–$10^{-11}$ dB/m, showing a relatively low CL. This is caused mainly by the high refractive index ring and the layer-by-layer amplification of the air hole structure. At the wavelength of 1550 nm, the CL of $EH_{16,1}$ mode is the lowest, which is $1.55 \times 10^{-11}$ dB/m, as low as the lower-order mode of a similar structure [11,12,14]; the CL of EH3,1 mode is $3.13 \times 10^{-10}$ dB/m, lower than the existing PCFs [35].

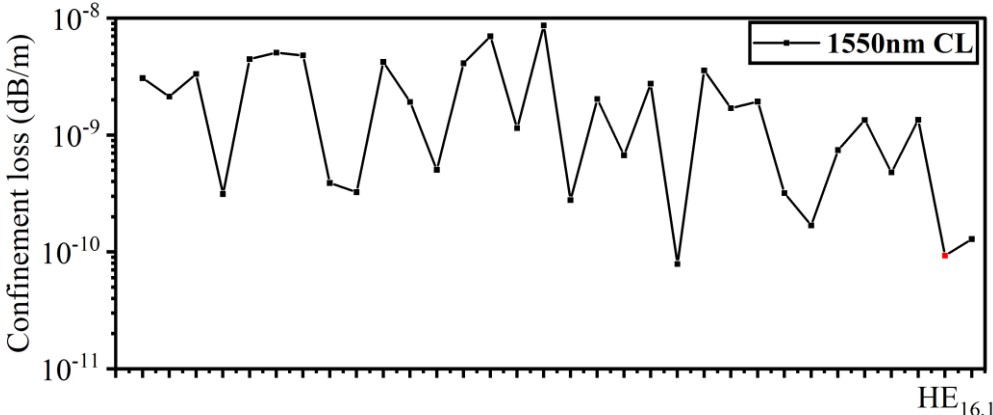

**Figure 9.** The distribution of CL for different vector modes at 1550 nm.

*3.5. Effective Mode Field Area (EMFA)*

The EMFA is used to represent the energy concentration of the mode. The smaller the EMFA, the more concentrated the energy. The quality of the mode field directly affects the NA and NC of the optical fiber, so the EMFA is an important research parameter in SC-PCF. *Aeff* can be used to express the EMFA of the OAM mode, and the formula can be used to compute it [36]:

$$A_{eff} = \frac{\left(\iint |E(x,y)|^2 dxdy\right)^2}{\iint |E(x,y)|^4 dxdy} \tag{6}$$

where E (*x,y*) represents the electric field distribution in the cross-section of the SC-PCF.

Figure 10 shows the relationship between *Aeff* and wavelength. It can be seen from the figure that *Aeff* tends to rise with the increase in wavelength. Through analysis of dispersion and CL, it indicates that the larger the wavelength, the weaker the limiting ability of the light field, and the energy of the fiber core leaks into the cladding gradually, leading to energy divergence, and *Aeff* increases gradually. It can be seen from Figure 10 that the relationship between *Aeff* and wavelength changes linearly. In the range from 1400 nm to 1700 nm, the overall distribution of *Aeff* is from 60 μm$^2$ to 110 μm$^2$. At the wavelength of 1550 nm, the $TM_{0,1}$ mode has the largest *Aeff* of 107.57 μm$^2$, higher than the current PCFs [31,35,37].

*3.6. Numerical Aperture (NA)*

NA represents the total amount of light energy in the fiber, which determines how much light energy the fiber collects, and this has important applications in the optical field. A higher NA is needed to improve the optical energy stored in the fiber core, when designing a fiber, which can be expressed as the ratio of *Aeff* to wavelength [31].

$$NA = \left[1 + \frac{\pi A_{eff}}{\lambda^2}\right]^{-\frac{1}{2}} \tag{7}$$

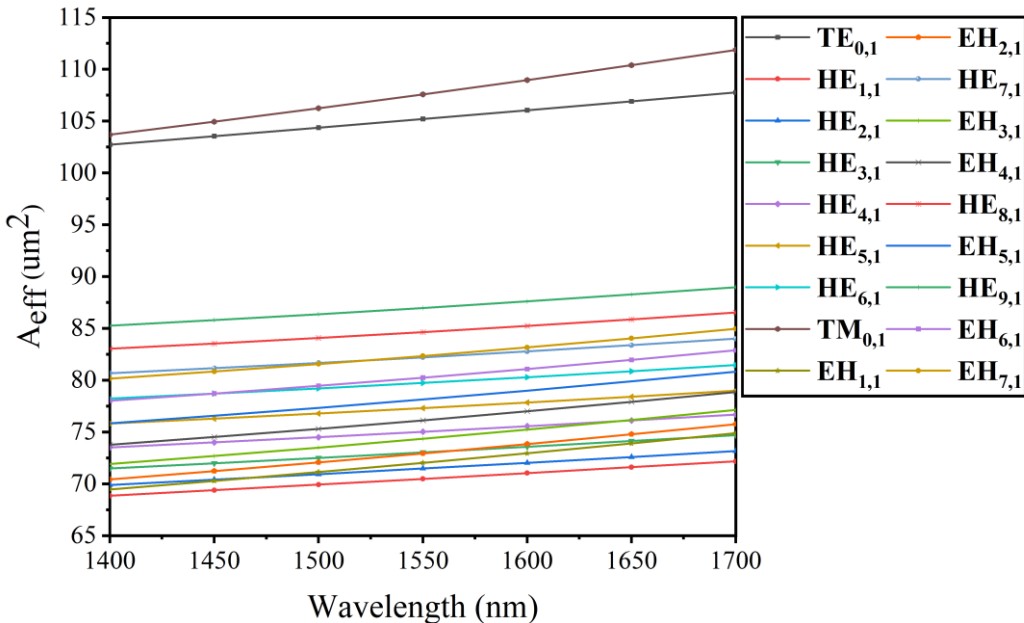

**Figure 10.** The relation between EMFA and wavelength in different vector modes.

Referring to Equation (7), NA can be increased by increasing the wavelength and decreasing *Aeff*. Figure 11 shows the changing relationship between wavelength and NA. The NA at 1550 nm is lower than 0.11, and the low NA is conducive to its application in the field of coherent layer analysis and imaging [38].

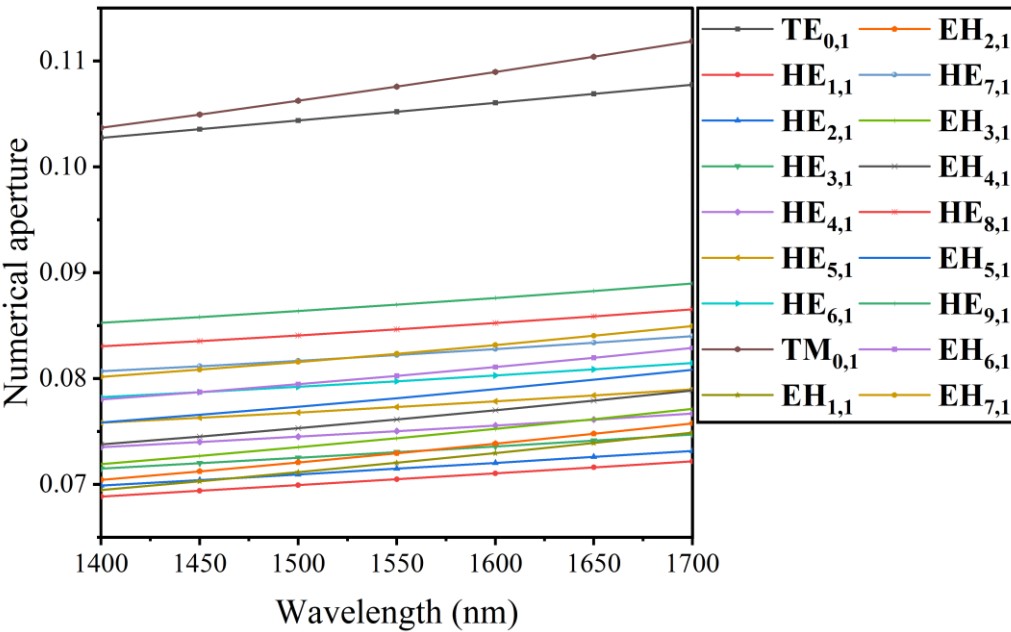

**Figure 11.** The relation between NA and wavelength in different vector modes.

*3.7. Nonlinearity Coefficient (NC)*

Nonlinearity is one of the most important optical characteristics in SC-PCF. Nonlinearity is usually represented by NC. The smaller NC, the weaker the nonlinearity, and the better the transmission efficiency of the information in the optical fiber. NC is represented by $\gamma$ [36]:

$$\gamma = \frac{2\pi n_2}{\lambda A_{eff}} \tag{8}$$

where $n_2 = 2.3 \times 10^{-20}$ m$^2$/W indicates silica's nonlinear refractive index. NC is proportional to $n_2$ and inversely proportional to *Aeff*, so that the NC decreases with the increase in EMFA.

Figure 12 shows the relationship between wavelength and NC. In the range from 1400 nm to 1700 nm, the overall NC is less than 1.5 W$^{-1}$·km$^{-1}$, showing extremely low NC, especially at 1700 nm; the lowest NC in TM$_{0,1}$ mode is 0.789 W$^{-1}$·km$^{-1}$. Optical communication mainly works in the C-band around 1550 nm, where the minimum NC of TM$_{0,1}$ mode is 0.867 W$^{-1}$·km$^{-1}$, which is 0.1 W$^{-1}$·km$^{-1}$ to 2 W$^{-1}$·km$^{-1}$ lower than the PCF structures proposed in recent years [18,21,35]. It is more conducive to the stable transmission of OAM mode in the optical fiber and has broad application prospects. For example, in optical fiber communication, suitable nonlinearity can be obtained by designing photonic crystal fibers with special structures, which can be used to make optical fiber communication devices, such as Raman fiber amplifiers, optical parametric amplifiers, etc. [39–41]; in optical soliton research, optical fibers can be used with nonlinear effects in the design of pulse compression or all-optical switches [42,43].

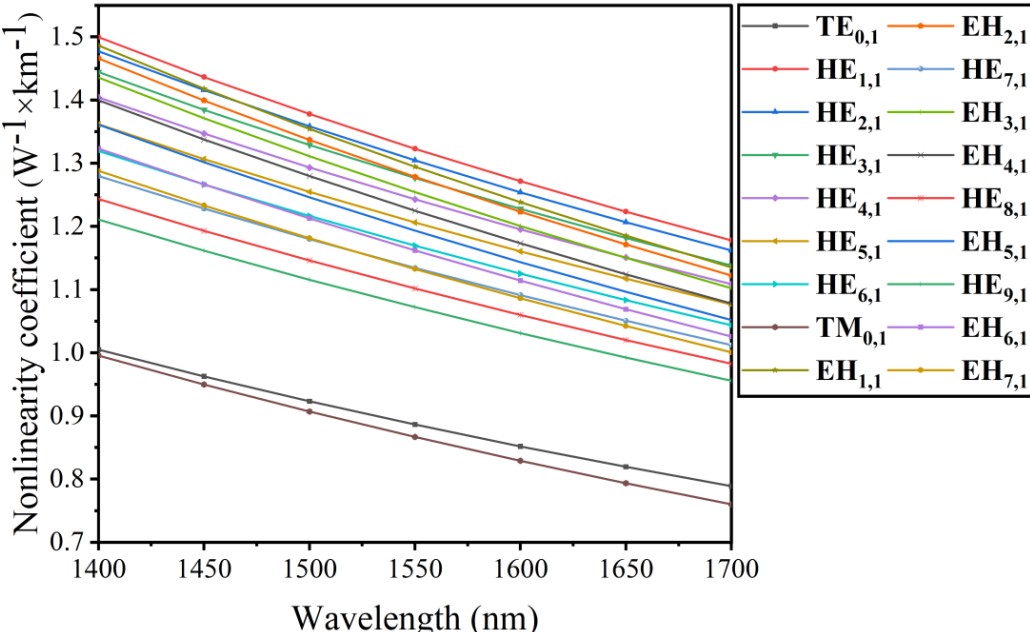

**Figure 12.** The relation between NC and wavelength in different vector modes.

### *3.8. Mode Purity*

Mode purity is also an important parameter to be considered in PCF design. High mode purity can ensure stable transmission and improve the transmission efficiency of OAM mode in the optical fiber. Mode purity can be expressed by the weight of mode intensity on the cross-section of the fiber [34].

$$\eta = \frac{I_r}{I_c} = \frac{\iint_{\text{ring}} |\vec{E}|^2 dxdy}{\iint_{\text{cross-section}} |\vec{E}|^2 dxdy} \tag{9}$$

where $I_r$ represents the average mode intensity in the annular region of the SC-PCF. $I_c$ represents the average mode intensity across the cross-section of the SC-PCF.

At 1550 nm, Figure 13 shows the mode purity for all modes supported by the SC-PCF. The mode purity of lower-order modes is higher than that of higher-order modes, as shown in the figure, and the values of mode purity for all modes are greater than 96%, with the mode purity of TE$_{0,1}$ mode reaching 97.9%. As a result, the optical fiber structure presented in this study can ensure the stable transmission of OAM mode and improve the information transmission efficiency at C-band.

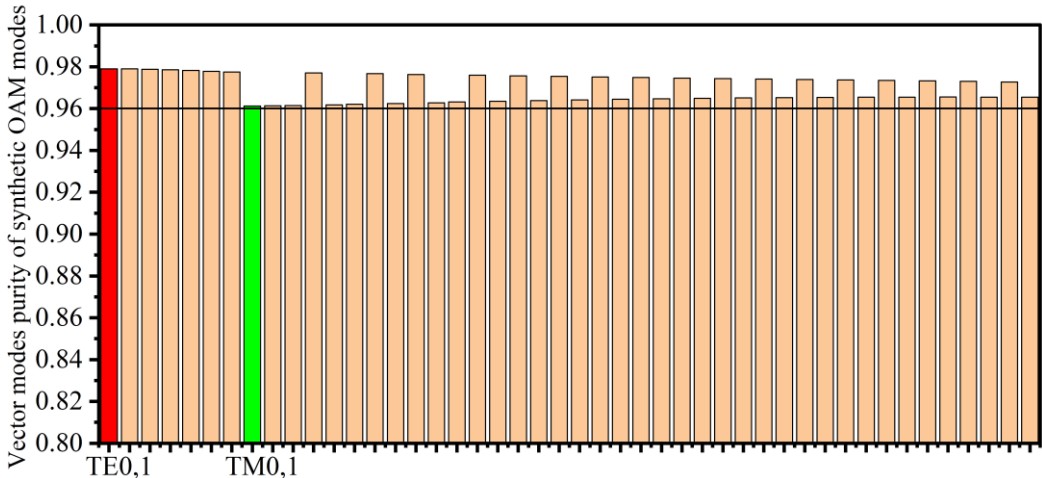

**Figure 13.** Mode purity of different vector modes at 1550 nm.

*3.9. Comparison with Existing Typical PCFs*

The SC-PCF proposed in this paper has been numerically calculated and its transmission characteristics have been analyzed. Table 3 lists the comparison of transmission characteristics between the SC-PCF proposed in this paper and the existing PCF without high refractive index material doping, including the number of OAM modes, dispersion at 1550 nm, CL, EMFA, NC, and mode purity. The SC-PCF supports 86 OAM modes, which is the maximum compared to the existing PCFs. The dispersion at 1550 nm of SC-PCF is from 4.939 ps/nm/km to 71.6 ps/nm/km, and CL is from $10^{-11}$ dB/m to $10^{-8}$ dB/m. It indicates that these two parameters are at extremely low levels. The large EMFA minimizes the NC of SC-PCF. At the same time, the mode purity of all modes is higher than 96%, which is at the highest level overall compared to the existing PCFs in the table. The comparison shows that the SC-PCF has excellent transmission characteristics. More modes can ensure that the optical fiber carries more information in the multiplexing system. Low dispersion, CL, NC, and high mode purity in the optical fiber are beneficial to improved OAM mode stability and long-distance transmission.

**Table 3.** OAM fibers based on PCF structures.

| Year | Number of OAM Modes | 1550 nm Dispersion (ps/nm/km) | Confinement Loss (dB/m) | Effective Mode Field Area (μm²) | Nonlinear Coefficient (w⁻¹·km⁻¹) | Mode Purity (%) | Bandwidth (nm) | Reference |
|------|------|------|------|------|------|------|------|------|
| 2016 | 26 | flat | <$10^{-9}$ | - | 356.2 ($HE_{3,1}$) | - | 1100–2000 | [44] |
| 2018 | 46 | >50 | <$10^{-9}$ | 54.25 ($HE_{3,1}$) | <2.58 | - | 1200–2000 | [16] |
| 2018 | 54 | >53.29 | <$10^{-9}$ | >48 | 1.41–2.1 | - | 1500–1600 | [17] |
| 2020 | 26 | <25 | <$10^{-8}$ | >35 | 2–3 | - | 800–1800 | [31] |
| 2020 | 38 | >4.75 | >$10^{-10}$ | - | >1.04 | >85 | 800–2000 | [30] |
| 2020 | 56 | >36.91 | $10^{-8}$ | >60 | <4 | <94 | 600–2500 | [45] |
| 2020 | 50 | >46.96 | <$10^{-9}$ | >80 | 0.6–1.5 | - | 1150–2000 | [18] |
| 2020 | 56 | - | >$10^{-8}$ | - | - | >80 | 1400–1700 | [46] |
| 2021 | 30 | >3.59 | <$10^{-8}$ | 30–50 | <4.144 | >90 | 1350–1800 | [37] |
| 2021 | 44 | >50 | >$10^{-10}$ | 40–65 | <3.36 | - | 1150–1800 | [47] |
| 2021 | 86 | >4.939 | <$10^{-9}$ | 60–110 | 0.867–1.5 | >96 | 1400–1700 | this paper |

## 4. Conclusions

In this paper, a novel PCF structure consisting of square and circular air holes is proposed. The SC-PCF is simulated using software based on the finite element method, and the transmission characteristics of the structure in the S + C + L band are analyzed. The structure enables 86 OAM modes in the wavelength range of 1400 nm to 1700 nm,

according to the research, and the ERID between the modes is as high as $5.88 \times 10^{-3}$, which avoids the coupling between adjacent vector modes into LP mode effectively. The minimum dispersion of the modes is 4.939 ps/nm/km, and the minimum change in dispersion is 0.956 ps/nm/km. The CL of all modes is less than $10^{-9}$ dB/m. At the same time, the EMFA ranges from 60 $um^2$ to 110 $um^2$, and the large EMFA causes the fiber to have extremely low NC ($0.867 \; W^{-1} \cdot km^{-1}$). The mode purity of all modes is greater than 96% at the wavelength of 1550 nm. According to these specifications, the SC-PCF has a flat dispersion, low CL, and high mode purity. Through the numerical analysis of the SC-PCF proposed, it supports a large number of OAM modes for stable transmission, has the characteristics of large capacity and low loss, and has a broad application prospect in multiplexing communication systems.

**Author Contributions:** Conceptualization, Y.Y. and Y.L.; data curation, Q.H. and L.X.; formal analysis, Y.Y.; funding acquisition, Q.H. and J.D.; project administration, J.D.; supervision, Y.L., Y.W. and Z.L.; writing—original draft, Y.Y.; writing—review and editing, Y.Y. and Y.L. All authors have read and agreed to the published version of the manuscript.

**Funding:** This research was funded by the National Natural Science Foundation of China (61905062, 61927815 and 62070506), China Postdoctoral Science Foundation (2020M670613), and Hebei Post-doctoral Scholarship Project (B2020003026). We acknowledge the support of the Key Laboratory of all optical networks and advanced communications networks of the Ministry of Education (Beijing Jiaotong University) (AON2019005).

**Institutional Review Board Statement:** Not applicable.

**Informed Consent Statement:** Not applicable.

**Data Availability Statement:** Not applicable.

**Conflicts of Interest:** The authors declare no conflict of interest.

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
