# Peer review of "Design of PCF Supporting 86 OAM Modes with High Mode Quality and Low Nonlinear Coefficient"

_photonics, doi:10.3390/photonics9040266_

Round 1

Reviewer 1 Report

See attached pdf

Reviewer 2 Report

The manuscript entitled „Design of PCF Supporting 86 OAM Modes with High Mode 2 Quality and Low Nonlinear Coefficient” presents the study of a novel type of fiber based on photonic crystals which enable multi-channel transmission of OAM modes of light within the range of 1400 – 1700 nm. The authors present a proposal of a design composed of silica with adjacent layers of circular and square holes with appropriately chosen diameters and lengths.

The main part of the paper is the numerical analysis with the use of the finite element method of the key properties of the proposed fiber and comparing them to the fibers already described in the literature. All of these parameters are discussed concerning the applications in optical communication.

The main advantages of the presented proposal are a larger number of supported OAM modes of large purity; low nonlinear coefficient and small confinement losses. It is also shown that the fiber has characteristics that enable the stable low-loss transmission of light. The ERIDS values indicate that the fiber can be adapted for multichannel transmission.

The paper is interesting, it is well written, and provides a novel proposal of optical fiber based on photonic crystals with high potential applicational use. In my opinion, it will be interesting for the readers of Photonics and I recommend the paper for publication.

However, I’m interested in whether the Authors considered the influence of the possible imperfections in the process of fiber manufacturing (cutting the holes). I’d like to see some comments on that problem.

To improve the quality of the presentation I’d consider changing Fig.9: maybe split it into two parts. Although the results of CL for various modes are plotted in different colors and with different point marks, it’s hardly readable.
